# Chemical Ecology of *Streptomyces*
*albidoflavus* Strain A10 Associated with Carpenter Ant *Camponotus vagus*

**DOI:** 10.3390/microorganisms8121948

**Published:** 2020-12-09

**Authors:** Anna A. Baranova, Alexey A. Chistov, Anton P. Tyurin, Igor A. Prokhorenko, Vladimir A. Korshun, Mikhail V. Biryukov, Vera A. Alferova, Yuliya V. Zakalyukina

**Affiliations:** 1Gause Institute of New Antibiotics, B. Pirogovskaya 11, 119021 Moscow, Russia; anjabaranowa@list.ru (A.A.B.); ap2rin@gmail.com (A.P.T.); prig67@mail.ru (I.A.P.); v-korshun@yandex.ru (V.A.K.); metrim@gmail.com (M.V.B.); 2Shemyakin-Ovchinnikov Institute of Bioorganic Chemistry, Miklukho-Maklaya 16/10, 117997 Moscow, Russia; dobr14@yandex.ru; 3Orekhovich Research Institute of Biomedical Chemistry, Pogodinskaya 10, 119121 Moscow, Russia; 4Department of Biology, Lomonosov Moscow State University, 119991 Moscow, Russia; 5Department of Soil Science, Lomonosov Moscow State University, 119991 Moscow, Russia

**Keywords:** antibiotics, antimycins, actinobacteria, *Streptomyces*, ants, *Camponotus*, antagonism

## Abstract

Antibiotics produced by symbiotic microorganisms were previously shown to be of crucial importance for ecological communities, including ants. Previous works on ant–actinobacteria symbiosis are mainly focused on farming ants, which use antifungal microbial secondary metabolites to control pathogens in their fungal gardens. In this work, we studied microorganisms associated with carpenter ant *Camponotus vagus*. Pronounced antifungal activity of isolated actinobacteria strain A10 was found to be facilitated by biosynthesis of the antimycin A complex, consisting of small hydrophobic depsipeptides with high antimicrobial and cytotoxic activity. The actinomycete strain A10 was identified as *Streptomyces albidoflavus*. We studied the antagonistic activity of strain A10 against several entomopathogenic microorganisms. The antifungal activity of this strain potentially indicates a defensive symbiosis with the host ant, producing antimycins to protect carpenter ants against infections. The nature of this ant-microbe association however remains to be established.

## 1. Introduction

Natural products are an abundant source of novel bioactive compounds. For the last four decades, only 33.3% of small molecules approved for clinical use by the FDA were purely synthetic compounds, whereas all other drug leads were of a natural origin in some way [1]. Recent studies on microorganisms from unusual environmental niches led to the discovery of several valuable bioactive compounds [2,3,4,5]. Moreover, recent studies on chemical ecology indicate that biosynthesis of antimicrobial secondary metabolites is a complex of genetically programmed responses to environmental signals, caused first and foremost by other organisms in a microbial community [6,7,8,9]. In other words, secondary metabolites serve as one of the ways of chemical communication between microorganisms [10,11]. Thus, studies of antimicrobial secondary metabolites in various ecological communities can both be harnessed for drug discovery and shed light on the molecular basis of inter-species interactions.

Multilateral communities that include organisms of various taxons (animals, plants, invertebrates, bacteria, and fungi) are of special interest. In these communities, symbiotic actinobacteria can defend their hosts by producing antimicrobial compounds. Recently described insect-microbial communities provide several such examples [12]. Many actinobacteria (*Streptomyces* spp., *Nocardia* spp., *Pseudonocardia* spp., *Amycolatopsis* spp. etc.) were found to form a strong ecological association with insects (ants, wasps, bugs, etc.) and defend their hosts, offspring, or food supplies against fungal or bacterial invasions [13,14,15,16].

Recent phylogenetic studies indicate that insect-actinobacteria associations are quite versatile: horizontal transfer and recruiting of new antagonistic strains from the environment are common ways of forming defensive symbiotic bacteria [14]. This tendency explains the similar spectrum of secondary metabolites produced by symbiotic actinobacteria associated with host organisms of very different origins [17].

Ants are one of the most successful terrestrial species, monopolizing on average 15–20% of terrestrial animal biomass [18]. Nonetheless, most publications devoted to ant symbiotic microorganisms and their antimicrobial compounds were mainly focused on leaf-cutting ants. These ants are included in the group of fungus-growing, or farming ants because they cultivate fungus for food. They were found to be a particularly useful model for studying insect-microbe symbioses due to the ant-fungus-actinomycete tripartite mutualism evolved by these ants to suppress the fungal pathogen *Escovopsis*, harmful to fungal gardens [19,20,21]. These findings point to a high biotechnological potential of ant-associated microorganisms, as well as a significant ecological impact of microbial secondary metabolites on complex ecosystems. Therefore, studies of less-explored ant species are of great interest. Previously, we studied actinobacteria associated with carpenter ants and isolated strains producing antimicrobial compounds nybomycin [22] and tetracenomycin X [23]. Here we report *Streptomyces* A10 strain producing antimycins and its activity against entomopathogenic fungi.

## 2. Materials and Methods

### 2.1. Collection and Microbial Isolation

The isolation of Streptomyces sp. strain A10 was carried out from adult workers of *Camponotus vagus* collected from a nest in Kasimovsky District, Ryazan region, Russia (55.01138 N, 41.73078 E). Bodies of five individuals were washed three times in sterile distilled water and then separately crushed by a tissue microhomogenizer in a sterile saline solution. Subsequently, 50 μL samples of the suspensions were spread on plates of M490 media (HiMediaLab) supplemented by nystatin (250 μg/mL) and nalidixic acid (10 μg/mL). After 21 days of aerobic incubation at 28 °C, the colonies were transferred, purified on oatmeal agar (International Streptomyces Project (ISP) 3) [24], and maintained as spore suspensions in glycerol (20 %, *v*/*v*) at −20 °C.

### 2.2. Phenotypic Characterization

Cultural characteristics of A10 were determined after incubation for 3 weeks at 28 °C on yeast extract-malt extract agar (ISP 2), oatmeal agar (ISP 3), inorganic salt-starch agar (ISP 4), glycerol-asparagine agar (ISP 5), peptone-yeast extract iron agar (ISP 6) and tyrosine agar (ISP 7) [24]. Spore chain morphology and spore ornamentation were examined by light microscopy (Fisherbrand AX-502, Thermo Fisher Scientific) and SEM (JSM-6380LA, JEOL) analyses of the culture grown on oatmeal agar for 14 days. Growth at different temperatures (10 °C and 40 °C) and at different NaCl concentrations (1%, 5%, and 8%) was determined on organic broth 79 [25] after 2 weeks of incubation. The utilization of sole carbon and nitrogen sources, the activity of enzymes, liquefaction of gelatin, and utilization of citrate and malonate sodium were studied using a SIB-test system (Microgen, Russia).

### 2.3. Gene Amplification and Phylogenetic Analysis

The extraction of genomic DNA of isolates was achieved using procedures described elsewhere [26]. For PCR amplification of the 16S rRNA gene, the primers 243F (5′-GGATGAGCCCGCGGCCTA-3′) and A3R (5′-CCAGCCCCACCTTCGAC-3′) were used. PCR reactions contained 1.5 ng/µL template DNA, 5 µL of ready-made polymerase reagent 5X ScreenMix (EvroGen, Russia), 0.1 µL of 100 µM of each primer, and sterile Milli-Q water to a final volume of 25 µL. The thermal cycling conditions comprised an initial denaturation step at 95 °C for 5 min, followed by 30 cycles of denaturation (94 °C for 45 s), primer annealing (68 °C for 120 s), and extension (72 °C for 90 s). A final extension step was performed at 72 °C for 10 min. PCR products were separated by electrophoresis on a 1% agarose gel and the corresponding bands were excised. The amplicons were purified and sequenced using a commercial service (EvroGen).

Amplification and sequencing of house-keeping genes—atpD (ATP synthase subunit beta), recA (recombinase A), rpoB (RNA polymerase subunit B), and trpB (tryptophan synthase subunit B)—was carried out as described previously [27].

The resultant sequences of the 16S rRNA gene were manually aligned against the corresponding sequences of representatives of the genus *Streptomyces* using MEGA X [28] software (https://www.megasoftware.net/). Phylogenetic trees were inferred using the Neighbor-Joining (NJ) and Maximum-Likelihood (ML) tree-making algorithms. The GenBank accession number for the 16S rRNA gene sequence of Streptomyces sp. strain A10 is MW165475.

Multilocus sequence analysis (MLSA) was carried out according to a previously described procedure [27]. Partial sequences of the four house-keeping genes of isolate A10 and most closely related *Streptomyces* species were trimmed to 431 bp (atpD), 468 bp (recA), 505 bp (rpoB), and 498 bp (trpB) and concatenated head-to-tail in-frame in the same order, generating a final sequence of 1,902 bp. Concatenated sequences were aligned using MUSCLE, and NJ and ML trees for each MLSA were constructed for the 16S rRNA gene as described above.

### 2.4. Fermentation and Antimicrobial Assay

Antifungal and antibacterial activities of Streptomyces sp. strain A10 were tested by Petri plate bioassay experiments against test microorganisms, including *Candida albicans* ATCC 14053, *Aspergillus niger* ATCC 16404, *Bacillus subtilis* ATCC 6633, *Staphylococcus aureus* ATCC 25923, as well as against entomopathogenic strains: *Bacillus thuringiensis* VKPM B-6650, *Beauveria bassiana* VKPM F-1357, *Conidiobolus coronatus* (*Entomophthora coronate)* VKPM F-1359, *Metarhizium rileyi* (*Nomuraea rileyi)* VKPM F-1360, *Ophiocordyceps sinensis* (*Cordyceps sinensis)* VKPM F-1479, *Conidiobolus coronatus* VKPM F-442, and *Lecanicillium lecanii* (*Verticillium lecanii)* VKPM F-837 (approved names are provided for entomopathogenic fungi, with taxons in parentheses and numbers in the VKPM collection).

For antibiotic production, organic broth 79 medium (200 mL in a 750-mL Erlenmeyer flask) was inoculated with 20 mL of the seed culture. Then the flasks were incubated at 28 °C with shaking (250 rpm) for 7 days.

The antagonistic activity of the *Streptomyces* sp. strain A10 against entomopathogenic fungi were determined by dual-culture assay [29] and cross streak assay [30] (Figure 1).

Dual-culture assay. A 5 mm diameter mycelial plug was cut from a 14-day-old fungal colony and transferred to the center of a PDA plate. Spores of antagonistic strain A10 were one-line streaked on PDA 30 mm away from the center. The fungi plate without strain A10 inoculation was used as a control. The dual culture plates were incubated at 28 °C for 7 and 14 days. Then, the colony radii of entomopathogenic fungi were measured to calculate the inhibition percentage using the following equation: mycelial growth inhibition $\left(\%\right)=\left[\frac{R_{1\;}-\;R_{2}}{R_{1}}\right]\times 100$, where R_1_ is the colony radius of entomopathogenic fungi in the control, and R_2_ is the colony radius of entomopathogenic fungi in dual culture plates. The *Streptomyces* A10 strain was tested in each test culture in three repetitions. Test cultures grown under the same conditions without strain A10 served as control.

Cross streak assay. The *Streptomyces* sp. strain A10 was cross streaked as a single line on solidified PDA media in a petri dish and incubated at 28 °C for 7 days. Test organisms were then cross streaked perpendicular to the original streak of isolates. The plates were incubated at 28 °C for 7 days. The zone of inhibition was measured for each microorganism. Control plates of the same medium without actinomycete growth were also simultaneously streaked with the test organism to study their normal growth. If the fungal pathogen is sensitive to the antibiotic-producing actinomycetes, then it will not grow near their colonies. The antagonistic activity was evaluated by the inhibition zone around the actinomycete isolates.

In vitro antimicrobial activity was determined using the broth microdilution method as suggested by the CLSI guidelines document M100-S25 [31,32,33].

### 2.5. Isolation and Identification of Antimycins

Fermentation broth of *Streptomyces* strain A10 was centrifuged to separate the mycelium. Culture liquid was extracted trice with *n*-butanol, residual antifungal activity (against *C. albicans*) was controlled by disc-diffusion assay. Biomass was extracted with ethanol under ultrasonic treatment for 50 min. The resulting extracts were combined and solvent was removed in vacuo, the residue was dissolved in 50% ethanol and separated by flash chromatography on C18-modified silica (Macherey-Nagel POLYGOPREP 100-50C18). The column (length 6 cm, ⌀ 4 cm) was filled by the sorbent in ethanol, the solvent was then displaced with water. The extract (2–3 g per run) was applied to the flash-column using the dry load approach. The column was eluted stepwise with a water–acetonitrile mixture (0%, 10%, 20%, 40%, 60%, 75%, 100%). The resulting fractions were analyzed by HPLC and their antibiotic properties were evaluated. HPLC analysis was performed on a 4.6 × 250 mm column (C18, Beckman, Ultrasphere, 5 μm), the column was equilibrated with 10% (*v*/*v*) acetonitrile/water mixture. The chromatography was conducted with a linear gradient of acetonitrile (10 → 95% for 20 min and isocratic elution with 95% for 10 min) at a flow rate of 1 mL/min and detection of absorbance at 205 nm. Final purification of the antimycin A complex was performed on an Interchim Puriflash 4250 preparative chromatograph using a 2.1 × 250 mm column (C18, Agilent Zorbax, 7 μm).

Chromatographic separation and on-line mass spectral analyses were performed with an Agilent 6340 Ion Trap equipped with an electrospray ionization (ESI) source coupled to an Agilent 1100 HPLC system (G1379A degasser, G1312A binary gradient pump, G1367B high-performance autosampler, G1316A column thermostat, and G1314A variable wavelength detector). The compounds were separated on a UPLC column (2.1 × 50 mm, YMC-Triart C18, 1.9 μm), maintained at 30 °C. Eluent A was triple-distilled water with LC/MS grade formic acid (0.1%, *v*/*v*), eluent B was LC/MS-grade acetonitrile with the same acid additive. The column was eluted at a flow rate of 0.35 mL/min: 0–10 min 100:0 → 0:100 (A:B, *v*/*v*); 10–12 min 0:100 → 100:0 (A:B, *v*/*v*). The detection wavelength was 205 nm.

The settings of the ESI source in positive ion mode were as follows: temperature, 335 °C; nebulizer pressure, 30 psi (N_2_); drying gas flow rate, 10 L/min (N_2_); capillary voltage, 3500 V. Acquisition parameters were as follows: skimmer, 40.0 V; cap exit, 128.5 V; Oct1 DC, 8.00 V; Oct 2 DC, 1.70 V; trap drive, 52.5; Oct RF, 187.1 Vpp; lens 1, (−5.0): lens 2, (−60.0 V); MS1 scan range, *m/z* 50–2200 Da (8100 Da/sec). Auto MS2 parameters were as follows: scan range, *m/z* 50–2200 Da (8100 Da/sec); number of precursor ions, 3; threshold MS1, 3 × 10^5^; fragmentation amplitude, 1.00 V. Samples were taken from solution in 200 μL of MeOH (HPLC grade). The injection volume was 2 μL.

## 3. Results

### 3.1. Isolation and Characterization of Streptomyces Strain A10 Associated with Carpenter Ants

The carpenter ants studied in this work were collected from a single nest located in the Ryazan region, Russia (Figure 2). Actinobacteria of *Camponotus vagus* isolated from five individual worker ants were tested separately and all colonies of mycelia-producing bacteria were selected due to their typical morphology. Results of 16S rRNA analysis showed that the majority of gene sequences corresponded to *Streptomyces* sp., singular strains were determined to belong to *Amycolatopsis*, *Kribella*, *Actinoalloteichus* spp.

In the inoculations from all the studied individuals, colonies of the morphotype of a strain that we have named A10 clearly dominated, accounting for at least 60% of the total number of mycelial prokaryote colonies growing on each of the agar plates. Alignment of its 16S rRNA sequence allowed us to compare it with a bacterial database (https://www.ezbiocloud.net/) and revealed that A10 belongs to genus *Streptomyces*. The closest typical strains are *Streptomyces albidoflavus* DSM 40455, *Streptomyces hydrogenans* NBRC 13475, *Streptomyces koyangensis* VK-A60, and *Streptomyces daghestanicus* NRRL B-5418 with 99.40% similarity level. For a more detailed phylogenetic analysis, other closely related strains were added to the comparison (ant-symbiotic strains isolated from leaf-cutting ant nests and known antimycin producers [17,34,35]) (Figure 3).

In the 16S rRNA gene-based neighbor-joining phylogenetic tree of 28 *Streptomyces* species, strain A10 was placed in the cluster with typical strains of *S. albidoflavus*, *S. hydrogenans*, *S. koyangensis*, *S. violascens*, *S. daghestanicus*, and also *S. fabea* and *S. cadmiisoli*, supported by a 95% bootstrap value (Figure 3). This phylogenetic approach does not always have sufficient resolution for definitive identification, therefore, now it is usually complemented by multilocus sequence analysis of house-keeping genes. The latter provides improved taxonomic resolution for the *Streptomyces* species [37]. The phylogenetic tree of concatenated sequences of the four protein-coding genes (the MLSA tree) for the studied isolate and closely related typical strains demonstrated a high consolidation of strains A10 and *S. albidoflavus* DSM 40455, supported by an 89% bootstrap value (Figure 4).

Recently, the *Streptomyces albidoflavus* taxon was expanded ten previously identified *Streptomyces* species were included into it based on 16S rRNA gene sequences, MLSA and DDH data, combined with similar cultural, micromorphological, and physiological attributes [38]. Despite strains *S. koyangensis* VK-A60 and *S. hydrogenans* NBRC 13475 being considered taxonomically separate from *Streptomyces albidoflavus*, they form a distinct clade, supported by a 99% bootstrap value [39].

The fact that the cluster that A10 belongs to (Figure 3) includes several strains isolated from leaf-cutting ants *Acromyrmex volcanus* (*Streptomyces* sp. Av25_1, *Streptomyces* sp. Av26_5, and *Streptomyces* sp. Av28_3), as well as a strain isolated from a fungus garden of *A. octospinosus* (*Streptomyces* sp. Ao10), and that all of them exhibited the ability to synthesize antifungal antimycins [17], suggests that genetically related actinobacteria may be involved in defensive symbioses due to their ability to synthesize similar metabolites encoded by related genetic clusters [40].

Using the polyphasic taxonomy approach, we studied cultural, morphological, and physiological characteristics of strain A10 (Table 1 and Table 2, Appendix A). The phenotypic properties of strain A10 were compared with those of streptomycetes forming the joint cluster on 16S trees, with preference given to species that have morphological properties similar to A10—spore surface and shape of sporophores (Figure 5, Appendix A).

Thus, based on high phylogenetic similarity and phenotypic resemblance to DSM 40455 type strain, the isolate A10 can be described as *S. albidoflavus.*

### 3.2. Antimicrobial Properties of Streptomyces sp. A10

The primary screening revealed that strain A10 has prominent antifungal activity against both yeast and fungi, whereas bacterial test microorganisms (both Gram-positive and Gram-negative bacteria) were unsusceptible to the studied isolate. Thus, this strain was selected for a thorough study of antifungal properties, including antagonistic activity against entomopathogenic strains.

Using dual-culture assay, we were able to detect that the *Streptomyces* sp. strain A10 inhibited the growth of three entomopathogenic fungi (Appendix A), however, in the dual-culture assay the growth of the colony of the two test cultures (*L. lecanii* F-837, *M. rileyi* F-1360) was uneven.

To clarify the antagonistic activity, we applied the cross streak method (Appendix A). The major drawback of the cross-streak method was the difficulty in obtaining quantitative data as the margins of the zone of inhibition are fuzzy and indistinct. Therefore, only qualitative information was obtained about antagonism of the *Streptomyces* sp. strain A10 against these two entomopathogenic strains (*V. lecanii* VKPM F-837, *N. rileyi* VKPM F-1360).

The results obtained from both quantitative and qualitative approaches are summarized in Table 3.

Antagonistic activity against Gram-positive bacteria *B. subtilis* ATCC 6633, *St. aureus* ATCC 25923, and the entomopathogenic strain *B. thuringiensis* VKPM B-6650 was absent when using a dual-culture assay and the modified cross streak method. There was also no activity against *A. niger* INA 00760.

### 3.3. Identification of Antifungal Compounds Produced by Strain A10

Strain A10 was cultured and extracted as described above. Since the most important problem was establishing the chemical nature of antifungal secondary metabolites, we performed activity-guided fractionation of the extracts, focusing mainly on the ability of the fractions to inhibit the growth of yeasts (*C. albicans*). HPLC analysis showed that the antifungal activity of the fractions is associated with a group of hydrophobic compounds that are significantly retained on the reverse-phase HPLC column (eluted from C18 with 90% percent of organic solvent) (Figure 6). HPLC chromatograms of purified fractions containing antimycins can be found in Appendix A.

The compounds were analyzed by LCMS with mass-fragmentation using the GNPS MS/MS spectra library [42], revealing the antifungal compounds to be components of the antimycin A complex. Antimycin-type depsipeptides are a family of natural products that share a common structural motif, consisting of a macrocyclic ring with amide linkage to a 3-formamidosalicylate unit. These compounds are produced by many actinomycete species and have significant structural diversity, including various ring sizes (there are 9-membered, 15-membered, and 18-membered antimycin-type depsipeptides) [43]. Even amongst more “classical” 9-membered depsipeptides, there is a number of structural variations. For example, the antimycin A complex includes antimycins A1–A20, bearing alkyl moieties in side chains (Figure 7).

Despite significant structural variations (Figure 7A), the antimycin A complex, as well as most antimycin-type depsipeptides [43], contain a conserved motif (depicted in blue on Figure 7A), yielding the characteristic fragmentation, including the corresponding fragment ion (Figure 7C) [34]. Interestingly, while analyzing the composition of the antimycin A complex isolated in this work, we observed a number of homologous compounds, along with substances with the same mass, but different retention times (Figure 7B). Despite the fact that homology can be explained by differences in the side chains (R^1^ and R^2^), we observed a different fragmentation pattern for some of the homologs (Figure 7D). Substances with the canonical fragmentation pattern tended to have a higher retention time than components with the same composition but the alternative fragmentation. The alternative fragmentation included an ion with *m/z* 279, previously detected in synthetic *O*-methylated antimycins [44], thus suggesting that these compounds also contain the altered conserved motif. The exact nature of the differences in conserved structural motifs in detected antimycin-type compounds cannot be established based solely on mass-fragmentation and requires further investigation.

## 4. Discussion

The *Camponotus vagus* ants (black carpenter ants) are widespread across Europe (north to south of Sweden and Finland), in the Caucasus, in northern Kazakhstan, Western Siberia to Altai, and also in northwest Africa. They usually inhabit the edges and clearings of deciduous forests or pine forests, with nests in old stumps and other woody remains [45]. Unlike termites, carpenter ants do not feed on wood but use it to build nests, carving out passages and chambers. In addition, nests usually have an underground part. *Camponotus vagus*, like other ants of this genus, consume live or dead insects that provide protein, but mainly feed on high-carbohydrate substrates, such as “honeydew” produced by aphids, floral nectar, sometimes honey and sweet fruits.

The humidity of the nests is usually carefully controlled by ants in nature. On the one hand, high humidity is necessary for the successful development of juveniles, on the other hand, it gives a chance for microorganisms, mainly fungi, to quickly colonize the chambers of the brood and the queen. Temperature increase can have the same effect: it both promotes the activity and growth of the ant colony and provokes rapid development of unwanted microorganisms inside the nest.

Such microorganisms can enter the nest in different ways. First, deadwood is colonized by various types of filamentous fungi, serving as major cellulose and lignin destructors in terrestrial ecosystems. Secondly, frequent contact of the ants with soil, which is one of the main natural pools of microbial diversity, promotes the adhesion of cells and spores of saprotrophic microbes to the cuticle of the ants. Thirdly, specific trophic strategies of the ants may lead them to contact with entomopathogenic species. In particular, carnivorous *C. vagus* collect and bring to the nest various insects, some of which can be infected. Moreover, living in a mutualistic relationship with aphids, which are frequently infected by fungal entomopathogens *Lecanicillium lecanii*, *Beauveria bassiana,* and *Metarhizium anisopliae* [46], may be a threat to carpenter ants.

Thus, the necessity of controlling unwanted fungal microorganisms in the nests could spur cooperation between carpenter ants and microorganisms capable of producing antimicrobial metabolites. Similar ecological communities are common amongst species associated with wood and plant residues [12,15,17,47]. Ant-associated microorganisms draw significant attention from both ecological and biotechnological points of view [17,20]. Nonetheless, ecological communities of carpenter ants are significantly less explored than those including leaf-cutting ants, however, our previous works [22,23] showed that they are a promising source of useful secondary metabolites.

In our study of *C. vagus* specimens, strain A10 was dominant among the mycelial actinobacteria. Strain A10 was attributed by the totality of phylogenetic and phenotypic characters to the species *Streptomyces albidoflavus* (Figure 3, Figure 4 and Figure 5, Table 1 and Table 2).

Primary screening of antibiotic activity against the yeast *Candida albicans* revealed a noticeable antifungal potential in this strain; therefore, its ability to suppress a number of entomopathogenic fungi was assessed (Table 3). Among the studied entomopathogenic fungi are *Conidiobolus coronatus* (also known by the name *Entomophthora coronata*), *Beauveria bassiana*, and *Metarhizium rileyi*, species that have a worldwide distribution and a wide range of hosts, including leaf-cutting ants *Acromyrmex octospinosus* [48], and which turned out to be highly sensitive to suppression by *Streptomyces albidoflavus* strain A10. We detected that entomopathogenic strain VKPM F-442 exhibits relatively high susceptibility to strain A10. Interestingly, the anamorph VKPM F-1359 demonstrated a lower inhibition rate (Table 3).

*Ophiocordyceps sinensis* (previously known as *Cordyceps sinensis* [49]) and *Lecanicillium lecanii* (the anamorphic form of *Akanthomyces lecanii*)—members of order Hypocreales—showed weakly and no inhibition, respectively, in a dual-culture test with A10 (Table 3).

Thus, based on high antagonistic activity against many undesirable fungi, strain A10 might be involved in putative protective insect–actinobacteria associations with carpenter ants, similar to those including leaf-cutting ants [13,17,19].

The observed antifungal properties of the *Streptomyces* isolate were studied at the molecular level. Extracts of culture broth were fractionated using bioassay-guided reverse-phase liquid chromatography. LC/MS/MS analysis of the active fractions (Figure 6) revealed characteristic fragmentation patterns of these antibiotics, unequivocally defining them as antimycins-type compounds. Despite most of the compounds corresponding to members of the antimycin A complex (Figure 7), some of the produced antibiotics display an unusual fragmentation pattern. This fact allows us to presume that some detected substances, despite having the same mass as known antimycins, differ in the conserved core fragment, thus forming a distinct subtype of antimycins. The exact nature of these differences will be further explored.

As shown in a recent work [40], antimycin biosynthesis gene clusters (BGCs) are widespread among various representatives of phylum Actinobacteria: among 1421 analyzed genomes, 73 antimycin BGCs were found, mainly in streptomycetes, but also in representatives of “rarer” genera, e.g., *Saccharopolyspora flava* DSM 44771, *Streptacidiphilus albus* NBRC 100918. According to the analysis of the architecture of antimycin BGCs, these gene clusters exist in four forms: short-form (S-form, 15 genes), intermediate-form (IP- or IQ-form, 16 genes), and long-form (L-form, 17 genes). Almost all strains with S-form antimycin BGCs studied by the authors represent a closely related clade (Clade I), all members of which are closely related to *S. albus* J1074. In fact, strain *S. albus* J1074 has been recently reclassified as *Streptomyces albidoflavus* [50]. Phylogenetic analyses of antimycin BGCs clearly demonstrate that vertical transmission is the primary way of dissemination of the gene clusters [40].

Not only the taxonomic but also the biogeographical distribution of actinobacteria capable of producing antimycin is worth noting. Putative antimycin producers have been isolated from a relatively large geographical area, including at least five of the seven continents: Africa, Asia, Europe, North America, and South America [40]. Among them were strains isolated from plants [29,51,52], sponges, and insects, suggesting that they may be involved in defensive symbioses. Such assumptions regularly find confirmation in reports on antibiotic-producing strains associated with higher organisms [6,13,14]. Population genetic analysis of *Streptomyces albidoflavus* shows that insect-associated strains are phylogenetically distant from soil and bottom-soil isolates, because insects provide isolated and more constant conditions for microbial inhabitants, with frequent interaction with the hosts, yet less disturbance from the surroundings [53].

## 5. Conclusions

The *Streptomyces* sp. strain A10 was isolated from black carpenter ants *Camponotus vagus* collected in the Central Russia region. The isolate was assigned to the species *S. albidoflavus* by phylogenetic, morphological, and biochemical characteristics. Strain A10 was found to exhibit pronounced antifungal properties, including activity against entomopathogenic fungi, especially against *Conidiobolus coronatus* and *Beauveria bassiana*. The antifungal substances were isolated and identified as antimycin-type compounds. Further work in this area, in particular, on the establishment of symbiotic relationships between the ants *Camponotus vagus* and the streptomycete *S. albidoflavus*, can make it possible to write a new page in the “book” of the chemical ecology of microbial communities associated with insects.

## Figures and Tables

**Figure 1 microorganisms-08-01948-f001:**
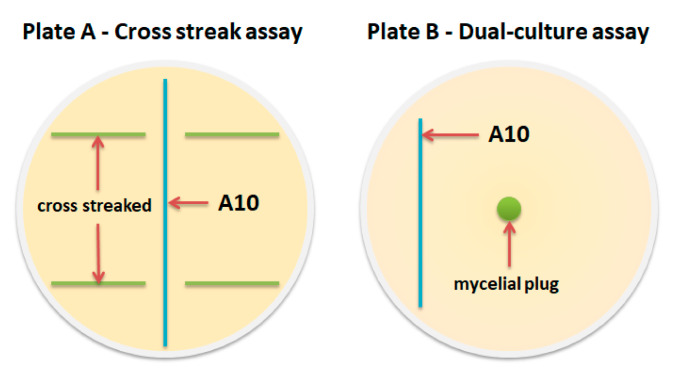
Illustration of the design of the used antagonistic studies: plate (**A**), Cross streak assay [30], plate (**B**), Dual-culture assay [29].

**Figure 2 microorganisms-08-01948-f002:**
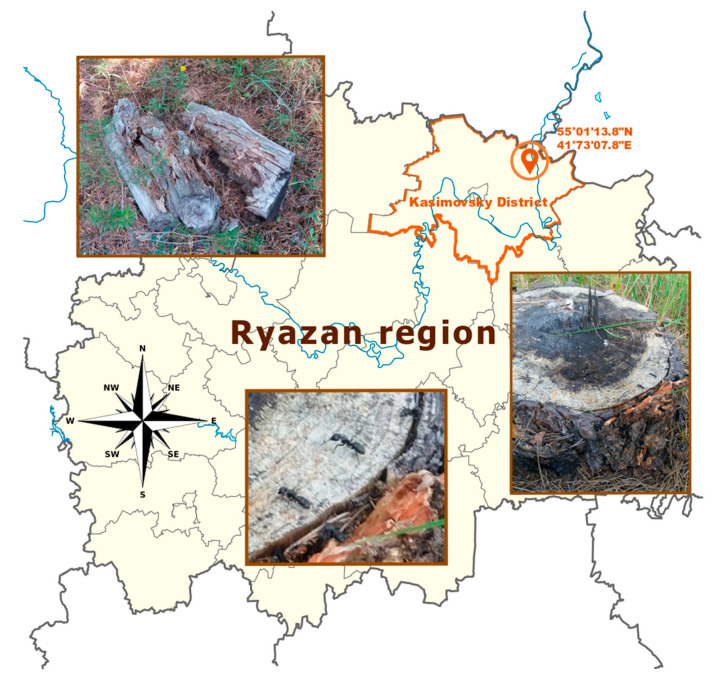
Region of collection of carpenter ants and nest photos.

**Figure 3 microorganisms-08-01948-f003:**
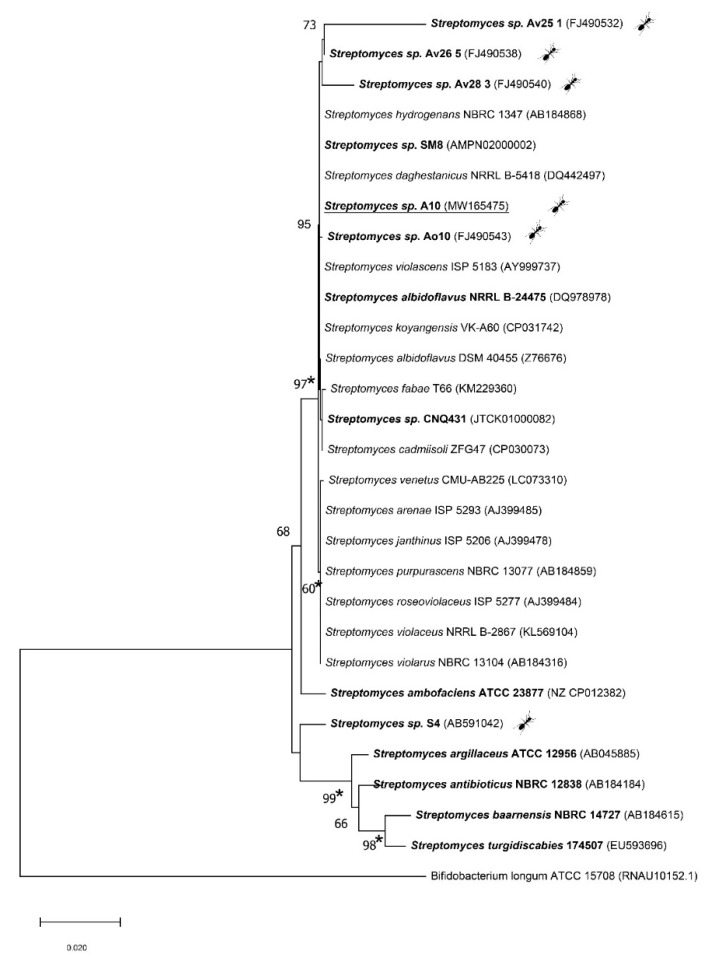
Neighbor-joining phylogenetic tree based on 16S rRNA gene sequence analysis showing the position of Streptomyces strain A10 (underlined) and related species. Antimycins-producing streptomycetes are highlighted in bold. Bootstrap values based on 1000 resampled datasets are given for values higher than 60%. *Bifidobacterium longum* ATCC 15,708 served as an outgroup. There were a total of 1164 positions in the final dataset. Asterisks indicate branches that were also obtained by the maximum-likelihood method [36]. The scale bar represents 0.02 substitutions per nucleotide position. Ant-marker indicates the strains, associated with leaf-cutting ants [17] and strain, isolated in this work.

**Figure 4 microorganisms-08-01948-f004:**
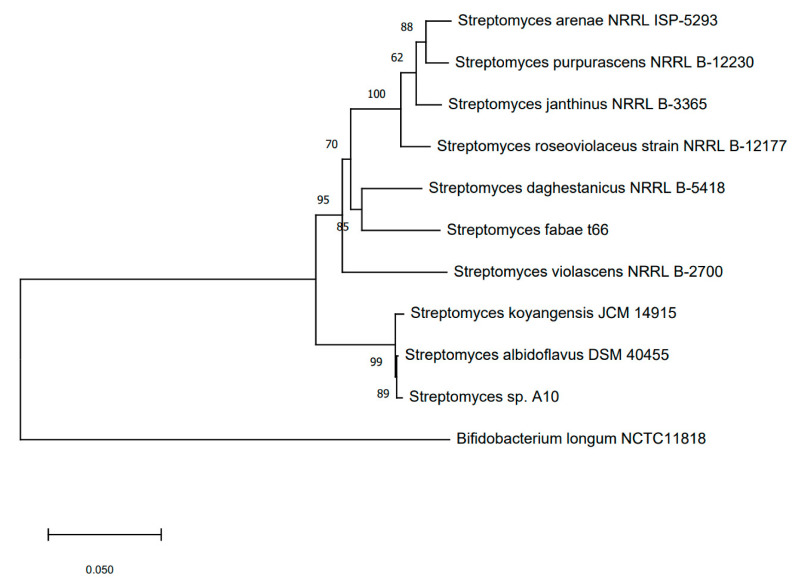
Neighbor-joining tree based on four concatenated sequences of house-keeping genes (MLSA tree), showing the relationships between strain A10 and typical strains of the *S. albidoflavus* and other related species. The evolutionary distances were computed using the Maximum Composite Likelihood method and are in the units of the number of base substitutions per site. The scale bar indicates 0.05 nucleotide substitutions per nucleotide position.

**Figure 5 microorganisms-08-01948-f005:**
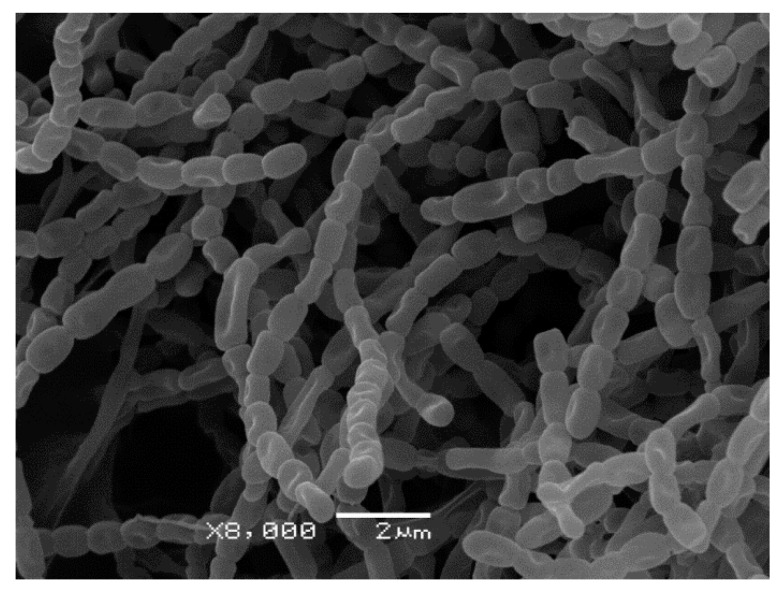
Scanning electron microscopy of strain A10 grown on ISP3 media for 8 days at 28 °C.

**Figure 6 microorganisms-08-01948-f006:**
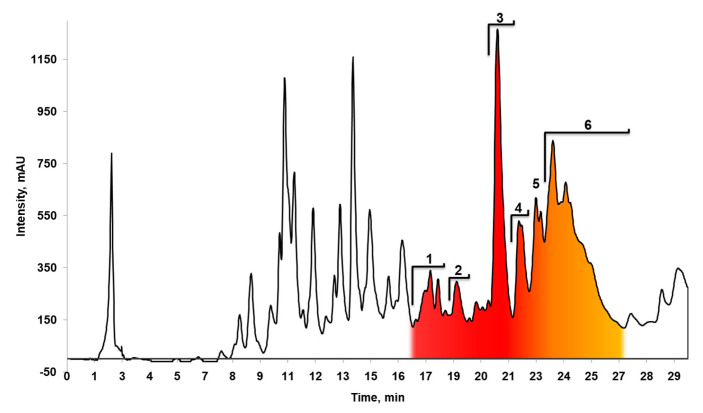
Analytical reverse-phase high-performance liquid chromatography (RP-HPLC) analysis of the antifungal fraction of *Streptomyces* sp. strain A10. Fractions 1–6 exhibited antifungal activity against *Candida albicans* and were collected for further analysis.

**Figure 7 microorganisms-08-01948-f007:**
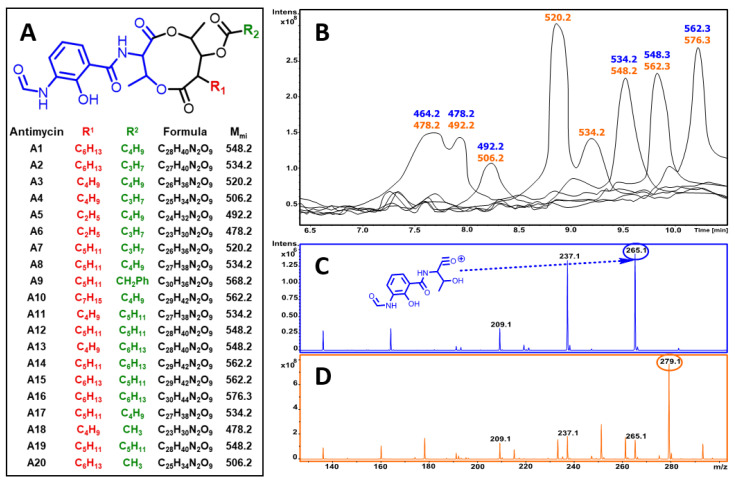
Structural identification of antimycins: (**A**) structures of antimycin A depsipeptides; (**B**) MS analysis of the obtained fractions, masses correspond to unprotonated molecules, the blue masses indicate the typical antimycin fragmentation pattern (**C**), the orange masses correspond to the 14 Da-shifted fragmentation pattern (**D**).

**Table 1 microorganisms-08-01948-t001:** Growth and phenotypic characteristics of *Streptomyces* sp. A10 and phylogenetically related Streptomyces species in various culture media *.

Strains	1	2	3	4
Yeast extract-malt extract agar (ISP-2)				
Growth	good	good	good	good
Aerial mycelium	greenish beige	sparse, white	white yellow	grey
Substrate mycelium	pale yellow	colorless	brown	ivory
Diffusible pigment	none	none	none	none
Oatmeal agar (ISP-3)				
Growth	good	good	good	good
Aerial mycelium	greenish beige	none	white yellow	grey
Substrate mycelium	light yellow	colorless	brown	ivory
Diffusible pigment	none/reddish	none/reddish	none	none
Inorganic salts-starch agar (ISP-4)				
Growth	moderate	good	good	good
Aerial mycelium	none	sparse, white	white yellow	grey
Substrate mycelium	pale yellow	colorless	brown	ivory
Diffusible pigment	none	none	none	none
Glycerol-asparagine agar (ISP-5)				
Growth	good	good	weak	good
Aerial mycelium	beige	white	white yellow	white/yellow
Substrate mycelium	yellowish white	colorless	creamy	yellow
Diffusible pigment	none	none	none	none
Peptone-yeast extract-iron agar (ISP-6)				
Growth	good	good	good	n/d
Aerial mycelium	slightly pink	white	white/grey	n/d
Substrate mycelium	brownish pink	brown	dark brown	n/d
Diffusible pigment	none	brown	dark brown	n/d
Tyrosine agar (ISP-7)				
Growth	good	good	good	n/d
Aerial mycelium	beige	sparse, white	white	n/d
Substrate mycelium	slightly pink	colorless	brown	n/d
Diffusible pigment	none	none	brown	n/d

* Standard media prepared according to the International Streptomyces Project methods [24]; strains/species: 1, *Streptomyces* isolate A10; 2, *S. albidoflavus* DSM 40455; 3, *S. koyangensis* VK-A60T; 4, *S. hydrogenans* NBRC 13475. Data for reference species 2 are from [38], for 3 from the website https://www.dsmz.de/collection/ and from [41].

**Table 2 microorganisms-08-01948-t002:** Morphological, physiological, and biochemical characteristics of strain A10 and phylogenetically related *Streptomyces* species.

Phenotypic Tests	1	2	3	4
Morphology spore chains	RF	RF	RF	RF
Spore surface	smooth	smooth	smooth	smooth
Melanoid pigment	-	-	+	-
Growth:				
1% NaCl	+	+	+	+
5% NaCl	+	+	+	+
8% NaCl	+	w	+	n/d
10 °C	+	n/d	+	n/d
45 °C	w	n/d	n/d	n/d
Acid from:				
Adonitol	-	n/d	-	n/d
Arabinose	-	-/+	+	+
Dulcit	-	n/d	n/d	n/d
Fructose	+	+	+	-
Galactose	+	+	n/d	n/d
Glucose	+	+	n/d	+
Inositol	-	-	-	-
Lactose	-	n/d	n/d	n/d
Maltose	+	n/d	n/d	n/d
Mannitol	+	-	+	-
Mannose	w	+	n/d	n/d
Raffinose	-	-	-	-
Rhamnose	-	-	-	-
Salicin	-	n/d	n/d	n/d
Sorbitol	-	n/d	n/d	n/d
Sucrose	w	w	-	-
Xylose	+	+	+	-
Enzymes:				
L-arginine	+	+	n/d	n/d
L-ornitin	+	+	n/d	n/d
L-lysin	+	+	n/d	n/d
Decomposition of:				
Cellulose	-	-	-	-
Gelatin	+	+	+	n/d
Starch	+	+	+	n/d
Urea	+	+	n/d	n/d
Production of:				
Soluble pigments	+	+	+	n/d
Citrate utilization	+	+	n/d	n/d
Malonate utilization	+	n/d	n/d	n/d
β-glucosidase	-	n/d	n/d	n/d

+, Positive; -, negative; w, weak; n/d, no data available.

**Table 3 microorganisms-08-01948-t003:** Antagonism of *Streptomyces* sp. strain A10.

Test Strain	Growth Inhibition Rate (%) * from Dual-Culture Assay	Growth Inhibition from Cross Streak Assay
*O. sinensis* VKPM F-1479	5.0 ± 3.2 ^b^	-
*C. coronatus* VKPM F-1359	15.7 ± 1.8 ^a^	+
*B. bassiana* VKPM F-1357	18.2 ± 3.6 ^b^	+
*C. coronatus* VKPM F-442	28.9 ± 3.4 ^a^	++
*M. rileyi* VKPM F-1360	n/a	+++
*L. lecanii* VKPM F-837	n/a	-
*C. albicans* CBS 8836	n/a	+++
*A. niger* INA 00760	n/a	-
*St. aureus* ATCC 25923	n/a	-
*B. subtilis* ATCC 6633	n/a	-
*B. thuringiensis* VKPM B-6650	n/a	-

* Growth inhibition rate (%) determined at 7 (a) and 14 (b) days after inoculation with entomopathogens on *Streptomyces* sp. strain A10. Data are means ± standard error over three replicates. n/a-test microorganisms did not form round colonies. -—no inhibition; +—weak inhibition; ++—moderate inhibition; +++—high inhibition.

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
