# Peer review of "Chemical Ecology of Streptomyces albidoflavus Strain A10 Associated with Carpenter Ant Camponotus vagus"

_microorganisms, 2020, doi:10.3390/microorganisms8121948_

Round 1
Reviewer 1 Report
In the manuscript, Baranova and colleagues report another observation that support: Streptomyces isolated from ants have the capacity to produce antifungal agents such as antimycins. While these compounds are not novel, the authors clearly highlight carpenter ants, in addition to leaf-cutting ones, may have ecological relationships with bacteria to reduce the fungal threat, especially from entomopathogenic fungi. The manuscript would be suitable in this journal; however, there are suggestions (see below) to improve its quality.
Broad comments/questions:
As there are previous studies that evaluate phylogenetic relationships of antimycin producers and its gene clusters (e.g. doi: 10.1099/mic.0.000572), please include: (i) a topic of discussion related to these studies, (ii) other studied antimycin producers (e.g. Streptomyces sp. S4 and S. argillaceus) in the phylogenetic analysis
Figure legends should stand alone (without the need to look for information put elsewhere in the manuscript). For example: while Figure 2 contains the analysis of 16S rRNA gene sequence (partial) as mentioned in the main text, but its figure legend does not have this information. Please check other figure legends.
The manuscript contains 9 figures. While the number of figures is not limited in this journal (I think), it would be worth to present the most important figures in the main part of manuscript while the others could be presented in a supplementary form. Figure 7 would complete Figure 5 and 6 but most images in Fig. 7 do not convey a clear message (and show heterogeneous growth?) – might be useful to present this data in different way?
If Figure 8 represent a chromatogram of the purified antifungal fraction, what are the early peaks in addition to peaks designated from 1 to 6?
The chromatogram shown in Figure 9B represents MS analysis of different fractions. Each of major peaks are identified with two masses with 14 Da differences, except for peaks 3. It would be worth to describe what these differences render to. Please also indicate partial structures represented by fragmentation masses 265.1 and 279.1 – perhaps in Fig. 9A
Appendix A: no description in the main text.
Clarification would be required for the methodology.
Section 2.5. Compounds were extracted from broth and mycelia separately, and later combined. What is the approximate ratio of antimycins isolated from broth to that from mycelia?
Line 157: What sorbent is used? Please also add the size of the column used.
Line 158: How much extract was applied?
Line 177: What is RF value used in MS?
Specific/additional comments:
Lines 148-150: Repetition of previous lines
Line 219: the 7 type strains – not clear
Line 310: Please explain what isobaric means here?
It might be useful to have a proofreading for language aspects:
- Paragraphs in the discussion section need to be reorganized
- Lines 326-327: do carpenter ants or fungi serve as cellulose and lignin destructors?
- Lines 376-379: confusing (need to rework)
Author Response
Referee #1:
In the manuscript, Baranova and colleagues report another observation that support: Streptomyces isolated from ants have the capacity to produce antifungal agents such as antimycins. While these compounds are not novel, the authors clearly highlight carpenter ants, in addition to leaf-cutting ones, may have ecological relationships with bacteria to reduce the fungal threat, especially from entomopathogenic fungi. The manuscript would be suitable in this journal; however, there are suggestions (see below) to improve its quality.
Broad comments/questions:
As there are previous studies that evaluate phylogenetic relationships of antimycin producers and its gene clusters (e.g. doi: 10.1099/mic.0.000572), please include: (i) a topic of discussion related to these studies, (ii) other studied antimycin producers (e.g. Streptomyces sp. S4 and S. argillaceus) in the phylogenetic analysis
Comparison with other studied antimycin producers was added to section 3.1 (Lines 209-229) and to the discussion (Lines 372-391). The phylogenetic tree was reworked (now Fig. 3).
Figure legends should stand alone (without the need to look for information put elsewhere in the manuscript). For example: while Figure 2 contains the analysis of 16S rRNA gene sequence (partial) as mentioned in the main text, but its figure legend does not have this information. Please check other figure legends.
The missing information was added to the caption of Figure 2 (now 3). Additional information was added also to the caption of Figure 5 (now 1).
The manuscript contains 9 figures. While the number of figures is not limited in this journal (I think), it would be worth to present the most important figures in the main part of manuscript while the others could be presented in a supplementary form. Figure 7 would complete Figure 5 and 6 but most images in Fig. 7 do not convey a clear message (and show heterogeneous growth?) – might be useful to present this data in different way?
Figure 5 was moved to the ‘Materials and methods’ section, and Figures 6 and 7 were moved to supplementary materials.
If Figure 8 represent a chromatogram of the purified antifungal fraction, what are the early peaks in addition to peaks designated from 1 to 6?
The caption of Figure 8 was confusing: this chromatogram is of an enriched fraction of the active extract containing all isolated antimycins, showing the position of antifungal compounds among other components. The caption of Figure 8 (now 6) was changed, chromatograms of purified fractions were added to supplementary materials (Figs. S3, S4).
The chromatogram shown in Figure 9B represents MS analysis of different fractions. Each of major peaks are identified with two masses with 14 Da differences, except for peaks 3. It would be worth to describe what these differences render to. Please also indicate partial structures represented by fragmentation masses 265.1 and 279.1 – perhaps in Fig. 9A
Structure of the conserved fragment with m/z 265.1 was added to Fig. 9C (now 7C). The exact nature of homology in the fragment with m/z 279.1 cannot be established based solely on mass-fragmentation and requires further investigation. The corresponding explanation was added to Lines 308-309, 312-313.
Appendix A: no description in the main text.
The figure in Appendix A (Fig. A1) was removed from the manuscript, the information on Maximum Likelihood method was included in reworked Fig. 2 (now 3).
Clarification would be required for the methodology.
Section 2.5. Compounds were extracted from broth and mycelia separately, and later combined. What is the approximate ratio of antimycins isolated from broth to that from mycelia?
At first we treated broth and mycelia separately. HPLC and in vitro bioactivity testing revealed similar antimycin content in both extracts, therefore, in the final procedures these extracts were mixed for fractionation.
Line 157: What sorbent is used? Please also add the size of the column used.
The size of the used column was added to Line 157. The sorbent is mentioned in the previous sentence: Macherey-Nagel POLYGOPREP 100-50C18.
Line 158: How much extract was applied?
About 2-3 g of the extract can be purified by this technique per run. The information was added to the manuscript (Line 157).
Line 177: What is RF value used in MS?
Extended MS parameters were added to the materials and methods section (Lines 175-181).
Specific/additional comments:
Lines 148-150: Repetition of previous lines
The repetition was removed in the mentioned lines.
Line 219: the 7 type strains – not clear
Text in the section 3.1 was rewritten, including Line 219 (now lines 216-217).
Line 310: Please explain what isobaric means here?
The term ‘isobaric’ was rephrased, because we cannot be sure that the compounds have a different composition (Lines 305-306, 369).
It might be useful to have a proofreading for language aspects:
- Paragraphs in the discussion section need to be reorganized
The discussion section was reworked (Lines 318-362).
- Lines 326-327: do carpenter ants or fungi serve as cellulose and lignin destructors?
The confusing sentence was rephrased (lines 328-336)
- Lines 376-379: confusing (need to rework)
The whole section of the discussion was expanded and reworked (Lines 372-391).
Reviewer 2 Report
In this study the authors set out to investigate potential ant associated microorganisms. They isolated a strain of actinobacteria which they designated Streptomyces strain A10 and characterised its phylogenetic position, antifungal and antibacterial potential as well as the likely identity of the antifungal compounds. They conclude that due to its the antifungal activity, strain A10 might be involved in a defensive symbiosis with the host ants.
While I liked the idea of characterizing potential ant associated microorganisms and to study their potential involvement in defensive symbiosis, I have several problems with the study.
The isolation of this strain might have nothing to do with a Camponotus association. Given that only five individuals were tested (line 74-75), presumably from the same nest, and given that whole though washed, animals were used, the presence of this strain could simply be a product of for example feeding on the same diet. While this does not diminish the characterization of strain A10, the study in my view needs to clearly state that any ant association is purely speculative at this point and inferred solely upon the basis of an antifungal activity of this strain. More ants from different colonies would need to be tested for the presence of this strain, to have a more solid basis to infer a potential ant association. In that respect, I also miss crucial information on abundance and frequency of this strain in the tested ants (see comment to Line 193). Finally, I found that the background given in the introduction, especially paragraph one (see comment to line 22.45) and the argumentation in the discussion are not well connected to each other and would benefit from rewriting.
Minor:
Line 33-45: This whole paragraph of the introduction describes the production of antibiotics of actinobacteria is dependent upon environmental conditions. While this information is interesting, I fail to see the connection to the objectives of the presented study, the characterisation of a novel Streptomyces strain in Camponotus vagus. At no point did the authors investigate whether the production of antibiotics of this strain in dependent upon environmental conditions.
Line 74-75: Were these five individuals taken from the same colony? The nest photo in Fig. 1 indicates only a single nest.
Line 119: I would suggest the authors remove the term clinically relevant for the microorganisms used to test antifungal and antibacterial activities of Streptomyces strain A10. While some of the used microorganisms definitely are most of the fungi are entomopathogens not clinically relevant for humans.
Line 125-127 and line 148 and 150: This sentence appears twice in these lines. Please correct.
Line 137: What does “The experiment was repeated twice with three replicates per treatment.” mean here in biological terms. Was only one isolated Streptomyces A10 strain tested twice with three replicates or were multiple isolated strains tested this way?
Line 193: What does “..most abundant strain (>60% of colonies)…” mean here. Does this mean that from all the colonies growing on agar plates inoculated with 50ul of the homogenized ant sample (line 76) over 60% were identified as strain A10? Does 60% refer to all plates from the five ant samples or is this kind of an average? How many colonies did grow overall on the plates inoculated with ant sample and how many of those were identified as strain A10 with 16S? What was the identity of other strains growing? More information is needed here.
Line 212: Does Fig. A1 refer here to Appendix A. Evolutionary analysis by Maximum Likelihood method? If so, please clearly label the figure in Appendix A as Figure A1.
Line 250-255: this information appears to me to be more appropriate for the materials and methods section if the authors screened potential antibacterial activity before testing different bacterial strains. On the other hand line 273-276 reports the results of antibacterial activity against some bacteria. I suggest the authors consider revising this section (header mentions antifungal activity but also results on antibacterial activity are reported). Moreover, in my view it would be good to report all results (tested fungi and bacteria) in Table 3 for a better overview.
Line 322: “Constanty contacting….” please consider revising
Line 341-360: This section with general information on the used fungi and the antifungal activity the isolated strain A10 had on them is in my view superfluous. Suffice to say that A10 had a variable activity against potential harmful fungi.
Fig. 5 appears to be more appropriate for materials and methods section.
Author Response
Referee #2:
In this study the authors set out to investigate potential ant associated microorganisms. They isolated a strain of actinobacteria which they designated Streptomyces strain A10 and characterised its phylogenetic position, antifungal and antibacterial potential as well as the likely identity of the antifungal compounds. They conclude that due to its the antifungal activity, strain A10 might be involved in a defensive symbiosis with the host ants.
While I liked the idea of characterizing potential ant associated microorganisms and to study their potential involvement in defensive symbiosis, I have several problems with the study.
The isolation of this strain might have nothing to do with a Camponotus association. Given that only five individuals were tested (line 74-75), presumably from the same nest, and given that whole though washed, animals were used, the presence of this strain could simply be a product of for example feeding on the same diet. While this does not diminish the characterization of strain A10, the study in my view needs to clearly state that any ant association is purely speculative at this point and inferred solely upon the basis of an antifungal activity of this strain. More ants from different colonies would need to be tested for the presence of this strain, to have a more solid basis to infer a potential ant association. In that respect, I also miss crucial information on abundance and frequency of this strain in the tested ants (see comment to Line 193). Finally, I found that the background given in the introduction, especially paragraph one (see comment to line 22.45) and the argumentation in the discussion are not well connected to each other and would benefit from rewriting.
Speculative nature of putative symbiosis between Camponotus vagus with A10 strain was emphasized in the discussion section (Lines 360-362). The rest of the remarks are commented on below.
Minor:
Line 33-45: This whole paragraph of the introduction describes the production of antibiotics of actinobacteria is dependent upon environmental conditions. While this information is interesting, I fail to see the connection to the objectives of the presented study, the characterisation of a novel Streptomyces strain in Camponotus vagus. At no point did the authors investigate whether the production of antibiotics of this strain in dependent upon environmental conditions.
The first paragraph in the introduction section was rewritten (Lines 33-43) to be less confusing.
Line 74-75: Were these five individuals taken from the same colony? The nest photo in Fig. 1 indicates only a single nest.
The tested individuals were taken from the same colony, information added to Lines 186-187.
Line 119: I would suggest the authors remove the term clinically relevant for the microorganisms used to test antifungal and antibacterial activities of Streptomyces strain A10. While some of the used microorganisms definitely are most of the fungi are entomopathogens not clinically relevant for humans.
The term ‘clinically relevant’ was removed from the sentence (Line 117).
Line 125-127 and line 148 and 150: This sentence appears twice in these lines. Please correct.
The duplicated sentence in lines 148 and 150 was removed.
Line 137: What does “The experiment was repeated twice with three replicates per treatment.” mean here in biological terms. Was only one isolated Streptomyces A10 strain tested twice with three replicates or were multiple isolated strains tested this way?
The clumsy sentence was rewritten (Lines 135, 136). There was only one Streptomyces A10, tested in three repetitions against each test strain.
Line 193: What does “..most abundant strain (>60% of colonies)…” mean here. Does this mean that from all the colonies growing on agar plates inoculated with 50ul of the homogenized ant sample (line 76) over 60% were identified as strain A10? Does 60% refer to all plates from the five ant samples or is this kind of an average? How many colonies did grow overall on the plates inoculated with ant sample and how many of those were identified as strain A10 with 16S? What was the identity of other strains growing? More information is needed here.
A more accurate description of the method was added to section 3.1, lines 186-200. 60% refers to the minimum number of colonies on each agar plate, some of them contained 100% of A10 colonies. We didn’t perform exact quantification of CFUs, because the results are highly dependent on a dilution, aliquot volume and numerous methodical errors. Some of the rest of isolated microorganisms were identified: Amycolatopsis, Kribella, Actinoalloteichus (lines 190-191).
Line 212: Does Fig. A1 refer here to Appendix A. Evolutionary analysis by Maximum Likelihood method? If so, please clearly label the figure in Appendix A as Figure A1.
Results of Maximum Likelihood method were added to reworked Fig. 2 (now 3), therefore Fig. A1 was removed from the manuscript.
Line 250-255: this information appears to me to be more appropriate for the materials and methods section if the authors screened potential antibacterial activity before testing different bacterial strains. On the other hand line 273-276 reports the results of antibacterial activity against some bacteria. I suggest the authors consider revising this section (header mentions antifungal activity but also results on antibacterial activity are reported). Moreover, in my view it would be good to report all results (tested fungi and bacteria) in Table 3 for a better overview.
The section 3.2 was renamed to ‘antimicrobial properties’ instead of ‘antifungal’. Figure 5 (now 1) was moved to the materials and methods section. The results of antibacterial testing were added to Table 3.
Line 322: “Constanty contacting….” please consider revising
The corresponding discussion section was revised completely.
Line 341-360: This section with general information on the used fungi and the antifungal activity the isolated strain A10 had on them is in my view superfluous. Suffice to say that A10 had a variable activity against potential harmful fungi.
The results of antifungal activity of the isolated strain A10 are now discussed briefly (lines 347-355).
Fig. 5 appears to be more appropriate for materials and methods section.
Figure 5 (now 1) was moved to the materials and methods section.
Round 2
Reviewer 1 Report
The authors have revised the manuscript substantially – Thank you very much for the effort! This includes the quality of presentation, the clarity of method and results. Now the manuscript is in a better shape for publication.
There is a super minor suggested edit:
Please add “the” in the sentence so it becomes “the sorbent” based on the authors’ reply below
Line 157: What sorbent is used? Please also add the size of the column used.
The size of the used column was added to Line 157. The sorbent is mentioned in the previous sentence: Macherey-Nagel POLYGOPREP 100-50C18.
Author Response
We added "the" as suggested (line 159).
Reviewer 2 Report
I am happy with the revision the authors have made. I have only a suggestion for the abstract which I hope that the authors will consider.
Line 27-28: Given the speculative nature of the association between the strain A10 and the ants, I would like to see this sentence in the abstract rephrased to: “The antifungal activity of this strain potentially indicates a defensive symbiosis with the host ant, producing antimycins to protect carpenter ants against infections”. Moreover I would appreciate it if the authors add to this sentence the following: “The nature of this ant-microbe association however remains to be established.”
Author Response
A couple of suggested sentences has been added to the Abstract section